# Linking Copper-Associated Signal Transduction Systems with Their Environment in Marine Bacteria

**DOI:** 10.3390/microorganisms11041012

**Published:** 2023-04-13

**Authors:** Pratima Gautam, Ivan Erill, Kathleen D. Cusick

**Affiliations:** Department of Biological Sciences, University of Maryland Baltimore County, Baltimore, MD 21250, USA

**Keywords:** marine bacteria, signal transduction system, copper, environment, comparative genomics

## Abstract

Copper is an essential trace element for living cells. However, copper can be potentially toxic for bacterial cells when it is present in excess amounts due to its redox potential. Due to its biocidal properties, copper is prevalent in marine systems due to its use in antifouling paints and as an algaecide. Thus, marine bacteria must possess means of sensing and responding to both high copper levels and those in which it is present at only typical trace metal levels. Bacteria harbor diverse regulatory mechanisms that respond to intracellular and extracellular copper and maintain copper homeostasis in cells. This review presents an overview of the copper-associated signal transduction systems in marine bacteria, including the copper efflux systems, detoxification, and chaperone mechanisms. We performed a comparative genomics study of the copper-regulatory signal transduction system on marine bacteria to examine the influence of the environment on the presence, abundance, and diversity of copper-associated signal transduction systems across representative phyla. Comparative analyses were performed among species isolated from sources, including seawater, sediment, biofilm, and marine pathogens. Overall, we observed many putative homologs of copper-associated signal transduction systems from various copper systems across marine bacteria. While the distribution of the regulatory components is mainly influenced by phylogeny, our analyses identified several intriguing trends: (1) Bacteria isolated from sediment and biofilm displayed an increased number of homolog hits to copper-associated signal transduction systems than those from seawater. (2) A large variability exists for hits to the putative alternate σ factor CorE hits across marine bacteria. (3) Species isolated from seawater and marine pathogens harbored fewer CorE homologs than those isolated from the sediment and biofilm.

## 1. Introduction

Copper is necessary in trace amounts as a biological ligand to support structural and catalytic activities within the bacterial cell [1]. Due to its redox properties, it functions as a cofactor for many enzymes involved in electron transport or redox reactions, such as cytochrome C oxidases, and superoxide dismutase [2,3]. Despite its role in cellular processes, the accumulation of excess copper in the cell is toxic, as it can lead to the production of hydroxyl radical that damages cellular macromolecules [4] and to the mismetallation of iron and manganese proteins [5] and displace metal cofactors from the active site(s) of enzymes, making them inactive [6]. These biocidal properties led it to serve as an early antimicrobial agent in various fields until commercial antibiotics were widely available [7,8].

Copper is becoming more prevalent in marine systems due to its use as an algaecide and as an antifouling (AF) agent on ship hulls following the ban on tributyltin (TBT) [9]. Due to its antimicrobial properties, copper is commonly used in AF coatings to inhibit bacterial growth and subsequent biofilm formation in systems, such as marine vessels, underwater pipes, and other surfaces, subject to biofouling/biofilm formation [10]. Thus, bacteria living in such environments have been exposed to a large amount of copper.

Bacteria rely on their various copper-regulatory systems to sense both intracellular and extracellular concentrations of copper. Different metalloregulatory or metal-sensing proteins monitor intracellular copper levels, which control the expression of genes necessary for excess copper homeostasis. These proteins form a co-ordinational complex with copper and allosterically activate or inhibit its binding with the DNA promoter of copper-regulatory genes [1]. Transmembrane sensor proteins primarily detect the extracellular copper levels in the two-component signal transduction system. They mediate copper homeostasis with the help of cognate response regulators, which control the expression of genes encoding copper detoxification systems in response to metal [11]. Extracytoplasmic function (ECF) σ factor can redirect RNA polymerase to initiate transcription from an alternate promoter and replace the primary σ factor in the presence of a suitable trigger [12]. All these systems must be regulated such that the metal sensor proteins are able to coordinate turning off copper uptake and upregulate the detoxification system when copper is deprived or present at toxic levels [11]. 

Marine bacteria exhibit significant biological diversity and can adapt to variations in environment and stress conditions, such as heavy metals, antibiotics, and toxins [13]. Many bacteria found in marine habitats have been found to harbor different sets of copper-regulatory mechanisms found either as a part of chromosome-encoded or plasmid-borne systems [14]. Bacterial resistance to different metals has been attributed to the presence of signal transduction systems that regulate genes belonging to different metal efflux systems [1]. In the marine environment, copper is prevalent for its use as an antimicrobial, but the regulatory mechanisms of copper in marine bacteria have not been extensively reviewed to date. Here, we provide an overview of the copper-associated signal transduction system in marine bacteria that senses the intracellular and extracellular copper status and regulates transcriptional machinery to maintain copper homeostasis. We begin by highlighting the current understanding of the known structural features of these regulators allowing them to bind the copper from the environment, leading to a transcriptional output response. We then evaluate the distribution of copper-associated signal transduction systems in marine organisms based on comparative genomic analyses of the copper-associated regulatory protein homologs. The use of comparative genomic analysis tools to analyze overall copper signal transduction systems on marine bacteria has not been previously explored.

The evolution of regulatory mechanisms to handle metal ion toxicity in various bacterial systems and how it is influenced by environmental pressures remain a topic that demands further exploration. We hypothesize that the distribution of signal transduction systems related to copper homeostasis in bacteria from the marine habitat is influenced by the environment.

### 1.1. Bacterial Signal Transduction System Overview 

Signal transduction systems allow bacteria to sense changes in extracellular and intracellular parameters and respond to changes by adaptive mechanisms [15]. They can broadly be defined as one- or two-component systems; additionally, the extracytoplasmic function (ECF) σ is also associated with copper signal transduction.

One-component signal transduction systems consist of a single protein capable of both sensing the environmental stimuli and affecting the adaptive response [16]. Most signal transduction in prokaryotes is predicted to be carried out by one-component signal transduction systems [16]. In this system, a single protein harbors two separate domains, the signal domain (input) and the response domain (output) [16]. The amino-terminal input domain is typically a small molecule (ligand) binding domain, and the carboxy-terminal output domain provides a site for DNA binding [17]. The output domain is in most cases a DNA-binding helix-turn-helix (HTH) domain that regulates gene expression through transcriptional activation or repression; it can also be a domain that mediates protein–protein interactions to regulate the activity of enzymes [16]. Although transcriptional regulators are not commonly referred to as one-component signal transduction systems, they harbor input and output domains similar to those found in two-component systems and carry out ligand binding and regulatory functions [16]. Some proteins are allosteric proteins that can directly bind to a specific metal ion and undergo a conformational change to allow control of the transcription of target genes [18]. One-component systems are predicted to detect cytoplasmic signals as evidenced by the lack of transmembrane region in most one-component regulators with DNA-binding domain [16]. 

The typical two-component signal transduction system consists of a sensor histidine kinase (HK) and a response regulator (RR), which are multidomain proteins. The HK comprises sensory and transmitter domains [19]. The RR has a receiver domain and an effector domain. In this system, the signals are sensed by an N-terminal sensor domain of the HK. The receiver domain of RR communicates with the transmitter domain of HK by phosphorylation, activating the effector domain of RR [11,19]. The phosphorylated form of RR initiates a corresponding cellular response primarily by transcription of the target gene [11]. 

Two-component systems are primarily known for detecting extracellular signals [11,20], although they might also detect cytoplasmic signals. Nearly 75% of sensor histidine kinases were predicted to contain one or more transmembrane regions and were thus predicted to detect extracellular stimuli [16]. However, the sensor domains may also be located within the membrane or entirely within the cytoplasm [21], where they detect intracellular signals. The C-terminal transmitter kinase domain of HK is cytoplasmic [22]. It is composed of a dimerization and histidine phosphotransfer (DHp) domain and CA (catalytic–ATP binding) domain [23]. The CA domain catalyzes the transfer of a phosphoryl group from ATP to the histidine residue [24]. The DHp domain contains the conserved histidine residue for the phosphorylation [24].

HKs may have more than one intracellular domain, which may consist of HAMP (found in histidine kinases, adenylyl cyclases, methyl-binding proteins, and phosphatases), PAS (Per-ARNT-Sim), or GAF (cGMP-specific phosphodiesterase, adenylyl cyclases, and FhlA) domains, along with the kinase domain with DHp and a CA domain [21]. 

RRs comprise an N-terminal receiver (REC) domain linked to a variable output or effector domain [22]. RRs harbor a phosphorylable aspartate residue within their conserved alpha–beta fold of the REC domain [25]. The phosphoacceptor aspartate catalyzes the phosphotransfer from the HK to its own regulatory domain and regulates the activity of the effector domain [24,26]. The effector domain is responsible for the output response with diverse effector domains in charge of different output responses, such as DNA binding or protein–protein interaction [24,27]. 

In addition to the one and two-component systems, extracytoplasmic function (ECF) σ factor is an important mechanism of bacterial signal transduction. Apart from the bacterial σ factor, which is a component of the RNA polymerase core enzyme, bacteria harbor alternate σ factors that can guide RNA polymerase to specific promoter sets supporting diverse functions [12,28]. The ECF σ factor belongs to a specific group of σ factor that contains σ2 and σ4 domains among the four conserved domains of σ factor [12,29]. The ECF σ factor is generally sequestered by a cognate anti-σ factor, which in the presence of a specific stimulus is inactivated by degradation or conformational change, resulting in the activation of the ECF σ factor [12]. The ECF σ factor related to copper homeostasis is CorE (Copper-regulated ECF σ factor), first identified in *Myxococcus xanthus*, which will be discussed in the later sections [30].

### 1.2. Signal Transduction Systems Associated with Copper

#### 1.2.1. One-Component Signal Transduction Systems of Copper Sensing 

Prokaryotic one-component signal transduction systems are abundant and diverse, classified into at least 20 families based on the amino acid conservation in their DNA-binding domain and defined by different conserved motifs [31]. Among them, some regulators are specific to metal ion homeostasis, classified based on the structural features of the metal-sensing coordination complex [18], and are named after the first identified member of the family [1,18]. 

The following section summarizes the major one-component systems involved in bacterial copper sensing and homeostasis of copper.

IMerR Family Regulators of Copper

*E. coli* encodes CueR, a one-component signal transduction system responsive to copper belonging to the MerR family regulator. MerR family regulators are an example of a one-component signal transduction system first named after the mercury resistance regulator MerR found in the Tn501 transposons [32,33]. The *E. coli* CueR protein is a regulator of the Cue copper efflux system under aerobic conditions [34,35]. In the presence of Cu(I), CueR activates the transcription of copper tolerance genes: P-type ATPase (*copA*), which removes Cu(I) from the cytoplasm to the periplasm, and the multicopper oxidase (*cueO*), which oxidizes Cu(I) to less toxic Cu(II) form in cytoplasm [8,36]. In addition to Cu, CueR activates the expression of the *copA* promoter in the presence of silver and gold [35,37]. Like other proteins regulated by MerR, the *copA* promoter has a long spacer region between the −10 and −35 elements [34].

CueR is also characterized in other organisms such as *Pseudomonas putida* PNL-MK25 and *Rhodobacter sphaeroides* [38,39], where it regulates either P-type copper efflux ATPase, copper chaperone or other copper-associated genes. In *P. aeruginosa*, the quorum-sensing global regulator LasR regulates the expression of CueR, which is involved in copper resistance [40]. *P. aeruginosa* CueR aligns closely with MerR family regulators, particularly copper-responsive CueR from *E. coli*, and with proteins that recognize monovalent cations such as ZntR and PbrR [40]. 

In the activator and repressor complexes of CueR and DNA, CueR is a dimer with each promoter contacting the dimer through the N-terminal DNA-binding domain (DBD), composed of four alpha helices in a winged helix-turn-helix motif, and C-terminal metal-binding domain [34,41]. In the CueR transcription activation complex (CueR-TAC), a topological switch of DNA makes the promoter suitable for RNA polymerase [42]. DNA-binding domain (DBD) of one of the two CueR subunits interacts with σ^70^ nonconserved region (σNCR) and plays a vital role in the bending of the promoter DNA which results in the switch from repressor complex to activator complex [41,43].

The overall mechanism of how CueR-induced transcription is turned off is not well understood. However, in *E. coli*, the CueR pool is maintained in the cell by ATP-dependent proteolysis by Lon and ClpP protease [44]. Cellular copper levels had little effect on the degradation of CueR, so degradation of CueR may operate continuously in *E. coli* [44]. This differs from copper-induced proteolysis observed in some species such as *Enterococcus hirae*, where expression of metallochaperone CopZ (regulated by the CopY repressor discussed later) increases up to 5 mM copper but declines at high concentrations [45].

IICopY Family Regulators of Copper

The one-component signal transduction system CopY was first identified in *Enterococcus hirae* [46]. In *E. hirae*, CopY regulates the transcription of the *copYZBA* operon, which encodes two copper-specific P-type ATPases (CopA and CopB), the copper chaperone CopZ [46] and homodimeric copper-inducible repressor (CopY). CopA and CopB are believed to be involved in copper uptake and efflux, respectively [47]. However, CopA might also drive cytoplasmic copper efflux but at a slower rate [48]. 

CopY family repressors involved in copper homeostasis have also been reported in other representative bacterial species, including *Streptococcus mutans*, *S. pneumoniae*, *Lactococcus lactis*, and *Enterococcus* species [46,49,50].

The CopY repressor is a bipartite protein with the N-terminal DNA-binding domain and the C-terminal containing a cysteine consensus motif found in copper and zinc-binding proteins [47,51]. At low copper concentrations, CopY is a Zn cofactor-bound homodimer *cop* operon promoter [52]. Under copper excess, copper-loaded chaperone CopZ transfers Cu(I) to the CopY repressor, which releases Zn (II) from the repressor. The release of one Zn(II) by 2 Cu(I) per CopY monomer leads to the derepression of *cop* operon by releasing promoter binding of the repressor [52,53]. A study on *Streptococcus pneumoniae* CopY suggests that CopY family repressors have a C-terminal CxC motif that binds to 2 Cu(I) or 1 Zn(II) [49]. The CopY-binding motif in *E. hirae* features an inverted repeat with consensus TACAnnTGTA, which is called a *cop* box. This region is conserved in CopY-like repressors from *Lactococcus lactis*, *Streptococcus mutans*, and other Firmicutes, and mutations in this sequence abolished the CopY–DNA interaction [50]. However, a recent study in *S. pneumoniae* suggests that its CopY repressor does not bind to the typical *cop* box sequence. This study suggests that the typical *cop* box sequence is necessary but insufficient for CopY binding. They proposed an updated *cop* operator sequence for *S. pneumoniae* [54]. 

A recent study has characterized a CopY family repressor PetR, along with a copper-responsive BlaR family membrane protease PetP, that regulates the switch between cytochrome c6 and plastocyanin in Cyanobacteria [55]. The membrane protease PetP regulates PetR levels in response to copper. PetR in this regulatory system represses *petE* (plastocyanin) and activates *petJ* (cytochrome c6) [55]. This regulatory system functions differently from other CopY family transcription regulators, where CopY regulator binds to copper directly [55]. In this case, PetR (CopY regulator) is degraded by the PetP in a copper-dependent mechanism, but whether this degradation also involves the binding of PetR to copper is not known. Although this system does not involve a downstream transcriptional response related to copper homeostasis like the other systems discussed in this review, its presence in cyanobacteria, major components of marine systems, necessitates its inclusion here. 

IIICsoR/RcnR Family Regulators of Cu

The CsoR (Copper-sensing operon repressor) family of metalloregulatory proteins is specific for copper sensing, unlike the MerR and CopY families, which respond to a wide range of ligands. The CsoR family is named after the representative protein of this family; CsoR was first studied in *M. tuberculosis* [56]. In *M. tuberculosis*, CsoR regulates copper-sensitive operon encoding *csoR* itself, a gene with unknown function, and a Cu(I)-efflux P-type ATPase *ctpV* [56].

CsoR homologs are primarily believed to be copper sensors in bacteria lacking other transcriptional regulators such as CueR and CopY [56]. *M. tuberculosis* also harbors a second CsoR homolog called RicR. RicR regulates a five-operon regulon encoding MymT, a copper-protective metallothionein; LpqS, a putative lipoprotein; Rv2963, a putative permease; the *socAB* (small ORF induced by copper A and B) operon; a putative multicopper oxidase; and the *ricR* gene itself [57,58]. *M. tuberculosis* harbors these two parallel copper-inducible pathways under the control of regulators CsoR and RicR, but how these systems differ in terms of their roles in *Mycobacterium* remains unknown. The CsoR mutant showed no significant change in the copper-responsive genes of the RicR regulon when exposed to copper stress but resulted in a hypoxia-type response, which might be the cellular response for disruption of copper homeostasis [59].

Other representative bacterial species found to harbor CsoR-like repressors include *B. subtilis*, *Thermus thermophilus* HB8, *Staphylococcus aureus*, *Listeria monocytogenes*, *Streptomyces lividans*, and *Corynebacterium glutamicum* [60,60,61,62,63]. 

CsoR from *M. tuberculosis* is an all-alpha helical protein and lacks the typical DNA-binding domain found in winged helix-type metalloregulators [56]. CsoR represses the transcription of the *cso* operon at low copper conditions by binding to the palindromic sequence GTAGCCCACCCC-N_4_-GGGGTGGGATAC [56]. 

CsoR in *M. tuberculosis* forms a homodimer and binds to two Cu(I) in a planar trigonal geometry via two cysteines and one histidine residue [51,56]. Other CsoR proteins studied in *Streptomyces lividans* and *Thermus thermophilus* form a homotetramer [61,64]. Most of the CsoR proteins have a C-H-C binding motif, but *T. thermophilus* has a C-H-H binding site for copper [61]. 

IVArsR/SmtB Family Regulator of Copper

ArsR/SmtB family is named after the first studied regulator of the family, arsenite/antimony sensor ArsR and *Synechococcus* PCC 7942 zinc sensor SmtB from *E. coli* [58,65]. This family includes homomeric metal sensor proteins involved in sensing various metal ions such as As, Sb, Bi, Zn, Cd, Pb, Co, Ni, Cu, and Ag. ArsR/SmtB family displays a great diversity with 13 metal-sensing and 2 nonmetal-sensing motifs characterized so far, where the metal ion binds [66]. In Cyanobacteria *Oscillatoria brevis* BxmR acts as a repressor of operon regulating the expression of ATPse metal transporter bxa, *bxmR* itself, and metal chaperone *bmtA* [56]. 

BxmR binds two Cu ions via cysteine residue, thereby causing conformational changes, which inhibit DNA binding of protein and activates the transcription [56]. The α3N and α5 sites of BxmR are the locations of the metal-binding site on its secondary structure [56]. BxmR binds to the DNA in a dimeric state and represses the operon under it. BxmR senses both Ag^+^/Cu^+^ and divalent metals Zn^II^/Cd^II^ [56].

VTetR Family Regulator of Copper

TetR family of one-component signal transduction system is mainly known for their roles as regulators of antibiotic efflux systems. It is a large family of regulators that respond to a wide variety of ligands, including the ComR regulator that responds to copper [31]. 

TetR-like transcriptional regulator ComR was first identified in *E. coli*. ComR is a repressor regulating the expression of the outer membrane protein ComC, which induces the membrane permeability to copper [67]. ComC is directly involved in controlling the copper leakage in the outer membrane, making the membrane less permeable [67]. Under copper-limiting conditions, *comC* is repressed by the regulator ComR. While in the presence of copper, ComR is released from the *comC* promoter site, and *comC* is expressed [67]. ComR-binding site was found at 60 bp upstream to the *comC* gene. ComR does not regulate its own expression like other regulators [67].

TetR family regulator is an alpha-helical protein with a large C-terminal domain and an N-terminal DNA-binding domain [68]. They typically act as a repressor and bind to the DNA, which upon ligand binding to the C-terminal domain, alter its binding to DNA [31]. 

Although the TetR-like regulator is widely distributed in bacteria, the copper-responsive ComR system is yet to be well characterized. It also remains unknown if ComR targets any other genes except the copper importer *comC*. ComC is also described as a stress-response protein showing its increased expression under several stress conditions [69]. ComC and ComR were initially described as YcfR and YcfQ and were linked to multiple stress responses [70,71]. 

#### 1.2.2. Two-Component Signal Transduction Systems of Copper Homeostasis

Bacterial copper homeostasis systems CusRS is a two-component signal transduction system that helps bacteria deal with excess copper in the environment. CusRS, first identified in *E. coli*, encodes chromosome-based histidine kinase CusS and a response-regulator CusR [72,73]. CusRS is required for copper-inducible expression of the *cusCFBA* operon that encodes genes required for copper efflux. CusA is a resistance nodulation and division (RND) family protein, a membrane-bound proton-driven transporter that exports different substrates from heavy metals to proteins. CusB is a membrane fusion protein, and CusC is an outer membrane fusion protein [74,75,76]. The CusCBA proteins form a multiunit transport complex that spans the membrane and the periplasmic space [34]. CusF is the copper-binding protein that interacts with the *cusCBA* gene for copper homeostasis [76]. In the plasmid-encoded copper resistance strains, CusRS also regulates the *pcoE* gene, a plasmid-borne periplasmic protein involved in the copper resistance [77]. It has been proposed that the Cus system is activated under the anaerobic condition when the CueO is inactive and induced when copper concentration overwhelms the one-component system Cue system [34]. The Cus system is also active in the presence of Ag(I). Silver ions can inactivate the multicopper oxidase CueO by binding to its Cu(I)-binding sites which disable the Cue system [78]. The Cus system was first characterized in *E. coli* for its role in detoxifying silver ions [76]. Later, the Cus system was found to be involved in the regulation of transcription of *cusCFBA* operon in response to both Cu(I) and Ag(I) [72,76]. 

CusS consists of a periplasmic sensor domain connected by a transmembrane helix to a catalytic cytoplasmic domain. The cytoplasmic domain comprises HAMP domain, DHp domain, and CA domain [11,79]. The CusR response regulator consists of a receiver and an effector domain. The CA domain catalyzes the autophosphorylation of a conserved histidine residue of the cytoplasmic domain. The phosphoryl group is then transferred to the aspartate residue of CusR in the receiver domain, which activates the transcription of cus operon [11,79].

The sensor domain of CusS binds to four metal ions (Ag or Cu ions) per CusS sensor domain and undergoes a conformational change upon the metal binding [80]. CusS is phosphorylated at the imidazole nitrogen N1 site at the conserved histidine [81]. The signal transduction mechanism in CusS involves autophosphorylation via the cis mechanism, contrary to what is known for the typical histidine kinases autophosphorylation [79]. The phosphorylated CusS transfers the phosphoryl group to aspartate residue D51 of CusR [79].

Studies have shown that RR CusR may have possible cross-talk with noncognate HK [82]. The copper tolerance was lower in CusR deletion mutant than the CusS-deleted strain, implying the possible role of cross-talk where noncognate HKs can also phosphorylate CusR in the absence of CusS [79]. 

The plasmid-borne copper resistance systems first found in *E. coli* and *Pseudomonas syringae* harbor the *pco* and *cop* operons with *pcoABCDE* and *copABCD* gene clusters, respectively [77,83]. These systems were first found in strains growing in excess copper environments that would overwhelm their chromosomally encoded copper systems [77,84]. The *pco* and *cop* loci are regulated by two-component signal transduction systems PcoRS and CopRS, respectively, which are required for the copper-inducible expression of copper resistance [77,84]. PcoS and CopS are homologous to sensor HK. They are predicted to have two cytoplasmic membrane-spanning domains with peptide loops extending to the periplasm [72]. When the concentration of copper is high, the kinases phosphorylate their respective response regulators, PcoR and CopR, converting them to transcriptional activators that induce the expression of their respective operons [72,84]. 

The system *pcoABCDE* codes for various inner, outer, and periplasmic proteins. The primary determinant of the system, *pcoA*, functions as a periplasmic multicopper oxidase which oxidizes Cu(I) to less toxic Cu(II) form [77,85]. The *pcoE* gene in pRJ1004 is not a part of the *pcoABCD* operon and is located further downstream on the plasmid and is controlled by its copper-regulated promoter CusRS [86]. In *E. coli* K12, the signal transduction systems are encoded by the *cus* operon and the *pco* operon, both play a role in maintaining copper homeostasis. However, they function as independent regulatory systems regulated by CusRS and PcoRS, respectively [76].

*Pseudomonas syringae* pathovar tomato harbors a homologous plasmid-borne copper resistance system known as *cop* system and encoded by six genes *copABCDRS* localized in the plasmid pPT23D [83]. The periplasmic CopA shares homology with the multicopper oxidase CueO (found in the chromosome), while CopB is an outer membrane protein whose function is poorly understood. The periplasmic protein CopC and inner membrane protein CopD are believed to be involved in the copper uptake system [87]. Some species, such as *Pseudomonas putida* KT2440, possess *copAB* but not *copCD*, and the corresponding operon is regulated by CopRS system [88]. 

A novel copper-responsive two-component DsbRS was discovered in *P. aeruginosa*, which induces transcription of the gene involved in the protein disulfide bond formation [89]. Copper binding to the DsbS histidine kinase inhibits its phosphatase activity leading to the activation of response regulator DsbR, which helps bacteria cope with copper stress. The DsbRS mutant is sensitive to copper, and its resistance was restored upon ectopic expression of the *dsbDEG* operon [89].

#### 1.2.3. Extracytoplasmic Function (ECF) σ Factor of Copper Homeostasis

ECF σ factors are a large group of ubiquitously distributed alternate σ factors in bacteria that are specialized in regulating the transcription of genes responding to different environmental conditions. The canonical ECF σ factor is regulated by a membrane-bound anti-σ factor, which binds to the σ-factor in the absence of a stimulus [28]. ECF factors are classified into many groups based on this diversity of regulatory functions and distinctive features in different studies [12,28]. A copper-responsive ECF σ factor, CorE (for copper-regulated σ ECF factor), was first identified in *Myxobacterium xanthus*. This system is involved in the copper homeostasis [30]. CorE-like σ factors are known to be regulated by their C-terminal extensions [12], which is different from the regulatory mechanism involving the anti-σ factor. 

In *M. xanthus*, CorE activates the expression of *cuoB* encoding MCO and *copB* gene encoding copper ATPase [30]. A second CorE homolog, CorE2, also identified in *M. xanthus*, is regulated by metals: cadmium and zinc [90]. 

CorE contains a cysteine-rich C-terminal extension involved in the binding of CorE to either Cu(II) or Cu(I) forms. Cu(II)-bound CorE leads to activation of the target promoter region. It is inactivated when bound to Cu(I) due to the reducing effect of the cytoplasm [30]. CorE ECF σ factor shares the CxC motif between the σ2 and σ4 subunit, which is also predicted to be involved in the coordination of metal [90]. CorE is predicted to be activated via a conformational change in the σ factor bound to the specific substrates, allowing promoter recognition and transcriptional response [90].

## 2. Materials and Methods

### Identification and Comparative Genomics of Copper-Associated Signal Transduction Protein Homologs

Copper-associated signal transduction putative protein homologs were identified in marine bacteria using the HMMER software package. The representative bacteria from the following phyla, Acidobacteriota, Bacteroidota, Calditrichaeota, Campylobacterota, Lentisphaerota, Planctomycetota, Actinomycetota, Bacillota, Pseudomonadota, and Cyanobacteriota, were used for comparative genomics. Bacteria used in the analyses belong to marine sources, including seawater, biofilm, sediment, marine pathogens, tidal flats, deep sea, and marine symbionts (Figure 1). In addition, the model organisms where the bacterial copper-regulatory system was identified were included in the analysis as a reference. 

The proteome sequences of 76 marine bacteria were downloaded from NCBI in FASTA format. The draft sequence was used for species when the complete genome was unavailable. The proteins associated with copper-associated signal transduction systems in bacteria were identified by literature review and were used as the proteins of interest for a homology search. Profile Hidden Markov models (HMM) of the proteins of interest were obtained from the PFAM and TIGRFAM, Panther (PTHR) databases (https://www.ncbi.nlm.nih.gov/protfam/ (accessed on 24 January 2023) & https://www.ebi.ac.uk/interpro/ (accessed on 24 January 2023) & http://www.pantherdb.org/ (accessed on 29 March 2023)), and a database of HMM of all models was compiled using the HMMER suite. The proteome sequence of organisms was scanned against the compiled HMM database to find protein homologs using hmmscan. The proteins of interest and their respective protein families are provided in Table 1. The search process was automated using a Python script. The hit count for different proteins was evaluated as a function of limiting e-value in 8 representative species to define an e-value threshold (Appendix A). Hits obtained with an e-value under 10^−30^ were used as sequence homologs for further comparative analyses.

The number of hits in each protein family representing the protein homologs for every organism was grouped to study the abundance of copper-associated signal transduction systems across marine bacteria. R studio was used for data visualization and analyses. The ggplot2 library was used for the graphical representation of the data. Protein homologs were first categorized based on the phylogenetic classification and then based on their isolation sources to predict the influence of the environment on the distribution of the copper-associated signaling systems. 

## 3. Results

### 3.1. Copper-Associated Signal Transduction System Homologs across Marine Bacteria and the Influence of Environment 

The putative copper-associated signal transduction system homologs in marine bacteria were identified using the database of HMM of proteins mapping to known signaling systems. The identified protein homologs were filtered based on an e-value threshold of 10^−30^. These protein homolog hits mapping to the reference TIGRFAM, PTHR, and PFAM family are referred to as counts in the Figures 3 and 4. We performed our comparative genomic analyses using representative marine bacterial species from the following phyla: Acidobacteriota, Bacteroidota, Calditrichaeota, Campylobacterota, Lentisphaerota, Planctomycetota, Actinomycetota, Bacillota, Pseudomonadota, and Cyanobacteriota. The comparative analysis of the signal transduction system associated with copper regulation among bacteria belonging to different phyla allowed us to identify any regulatory features shared by all members from a phylum. We considered the impact of the environment on the distribution of protein homologs by classifying them further based on the marine source of isolation. 

A subset of species from the overall comparative analyses was used for the 16S phylogenetic analyses to visualize the overall copper-associated systems across all phyla. When 16S sequence information was unavailable, we used a close relative of the strain for their 16S rRNA phylogenetic alignment. The distribution, as the presence/absence of copper signaling system homologs among the phylogenetic groups, was visualized with the 16S rRNA tree superimposed by the regulator information in Figure 2.

Our analyses show that diverse putative copper-associated signaling systems are widely distributed among marine species. The two-component CusRS system homologs and the ECF σ factor CorE system homologs were identified in all the species (Figure 2). The CusRS model used in our analyses includes homologs of the two-component CusRS system and the plasmid-borne two-component CopRS and PcoRS systems, as they all matched the same protein family model. The abundance of two-component system homologs in all marine species indicates that efflux and detoxification are the primary mechanisms of copper homeostasis in most species. The presence of ECF σ CorE homologs in all the bacteria strongly suggests the role of this system as a mechanism of copper-associated stress regulation. Our analyses demonstrate that some species possessed larger homolog hits to a specific copper system than other species, although they belong to the same phyla.

#### 3.1.1. Pseudomonadota Phylum

Pseudomonadota represented the largest group of marine bacteria used in the analyses. Species belonging to Pseudomonadota were isolated from sources, including seawater, sediment, biofilm, deep sea, pathogen, symbiont, copper models, and tidal flats.

The overall distribution of signal transduction systems suggests that most bacteria from the Pseudomonadota phylum possessed CueR one-component system homologs, which are not found in other phyla (Figure 2). Pseudomonadota showed significant differences in the putative CueR homolog count between different species (Figure 3). The species isolated from sediment and biofilm samples showed a large homolog hit compared to other organisms isolated from sea ice and saltern suggesting that the environment may influence this diversity. *Marinobacter psychrophilus* and *Saliniradius amylolyticus*, isolated from sea ice and saltern sediment, respectively, showed no CueR system hits. Similarly, *Marinobacter* sp., isolated from a self-regenerating biocathode biofilm, is the only species with seven homologs of CueR and *Marinovum algicola*, isolated as a symbiont of the toxin-producing dinoflagellate, harbored six CueR system hits. Bacteria isolated from the sediment, such as *Hydrocarboniclastica marina*, *Glaciecola* sp., and *Salinimonas sediminis*, harbored four CueR hits, unlike other marine species with 1, 2, or 3 CueR homologs (Figure 4). Microorganisms in the marine sediment play an important role in the geochemical processes in marine ecosystems [91]. This could be related to the presence of multiple hits to one-component copper systems when compared to organisms from other sources.

Pseudomonadota does not harbor other copper-associated one-component system homologs except for some *Alteromonas* sp., which showed the presence of CopY homologs (Figure 2).

We observed differences in the CusRS hits between the different species from phylum Pseudomonadota. *Croceicoccus naphthovorans*, *Roseobacter denitrificans*, *Candidatus Puniceispirillum marinum*, *Ruegeria* sp., and *Marinovum algicola*, belonging to Alphaproteobacterial class of Pseudomonadota, have much lower hits (4–12 counts) for the CusRS system (Figure 3). The species belonging to the Gammaproteobacteria class of Pseudomonadota also showed variability within the total CusRS homologs hits across species. Regarding environmental influence, *Saliniradius amylolyticus*, isolated from solar saltern sediment, showed five CusS and six CusR homolog hits. The metal ion-reducing bacterium *Shewanella oneidensis* MR-1 and a marine symbiont *Agarivorans gilvus* harbor fewer CusS homologs (11 and 8 CusS homologs, respectively). In comparison, species isolated from the biofilm, such as *Catenovulum* sp. and *Alteromonas* sp. RKMC-009, harbored many putative CusS homologs (29 and 28 CusS homologs, respectively) (Figure 3 and Figure 4). Overall, although the number of putative CusRS homologs was variable in Pseudomonadota, we observed no explicit difference based on the influence of the environment on this distribution.

All bacteria from Pseudomonadota harbored CorE ECF σ factor protein homolog hits in numbers ranging from five to eight counts (Figure 3). 

Similarly, *Marinovum algicola* is the only species from Pseudomonadota that harbors a single ComR homolog. This strain of *Marinovum algicola* contains 11 large extrachromosomal replicons in its genome and has been isolated as a symbiont of toxic dinoflagellate cultures. The presence of a multipartite genome might relate to the presence of multiple copper-regulatory system hits.

#### 3.1.2. Cyanobacteriota Phylum

Among the representative bacteria belonging to the phylum Cyanobacteriota, all bacteria harbor the two-component CusRS system and the CorE system homologs but not one-component copper system homologs (Figure 3). These organisms were found in seawater, marine sediment, and thermal spring (grouped with deep-sea organisms) (Figure 4). Among the representative species used for the study, *Prochlorococcus* sp., *Anabaena* sp., and *Synechococcus* sp. harbor 10, 30, and 22 total homologs of copper-responsive signal transduction system, while *Halomicronema hongdechloris* displayed 47 total homolog hits for these systems (Figure 3). It is intriguing to note the niche differences, as *Prochlorococcus* sp. and *Synechococcus* sp. are typically found throughout the water column, displaying more of a planktonic lifestyle, while *H. hongdechloris* is associated with stromatolites. *Trichormus variabilis* isolated from a thermal spring harbored 48 CusS homologs which might be related to the rich mineral system and overall metal abundance in this habitat [92]. Although we observed differences in the copper system homologs between the different environments analyzed here, a larger representative sample would be crucial to understand the distribution of copper systems in Cyanobacteriota.

#### 3.1.3. Other Phyla

Our analyses indicate that the other marine bacteria belonging to phylum Acidobacteriota, Bacteroidota, Calditrichaeota, Campylobacterota, Cyanobacteriota, Lentisphaerota, and Planctomycetota only harbored the two-component system homologs and ECF σ factor homologs (Figure 2). The Calditrichaeota *Caldithrix abyssi*, isolated from a deep-sea hydrothermal vent, showed 43 hits to copper systems. In comparison, bacteria from Acidobacteriota and Bacteroidota phyla isolated from sediment and seawater, respectively, harbored 9–12 hits (Figure 4). Similarly, bacteria from Campylobacterota phylum isolated from the deep sea, and bacteria belonging to Lentisphaerota, and Planctomycetota, isolated from the seawater, harbored 15–23 total hits to CorE and CusRS system homologs (Figure 4). The representative marine bacteria from these classes were comparatively fewer than those obtained for the other phyla and could be explored on a larger scale to characterize putative copper-associated signal transduction systems in these groups.

#### 3.1.4. Actinomycetota Phylum

All bacteria from phylum Actinomycetota harbored the one-component system CsoR homologs (Figure 2). Species belonging to this group showed one to two hits to the copper-sensing CsoR system. 

Similarly, three species from Actinomycetota harbored ComR one-component system homolog (Figure 2). ComR is a copper sensor that modulates the expression of genes encoding membrane proteins affecting the membrane permeability to copper [67]. The three strains with ComR homologs of Actinomycetota phylum were classified as marine fish pathogens. ComR system homolog was not observed in the nonpathogenic strains from the same group (Figure 3 and Figure 4). ComR binds to copper ions and acts as a repressor to regulate genes that affect the import of copper into cells. The presence of the ComR homolog in pathogenic strains shows a potential bacterial response against copper-associated toxicity from the host defense system [93].

Bacteria belonging to the Actinomycetota showed multiple homolog hits to the CusRS system. In this phylum, CusR hits were found in larger numbers (6–22 hits) than hits to the histidine kinase CusS (5–17 hits). This is opposite to what was observed in Pseudomonadota and in line with the results observed for Lentisphaerota and Planctomycetota. Similarly, one to five hits to the CorE system were identified in Actinomycetota species (Figure 3). Species from this phylum (*Renibacterium salmoninarum*, *Serinicoccus marinus*, and *Salinibacterium amurskyense*) harbored fewer overall copper regulatory systems than other organisms from the same phylum used in the analyses (Figure 3). These species were isolated as either pathogens of marine invertebrates or found as seawater species (Figure 4). The pathogenic and seawater species may undergo lesser environmental fluctuations and thus harbor fewer copper system homolog hits.

#### 3.1.5. Bacillota Phylum

Bacterial species belonging to the Bacillota phylum harbored either CsoR or CopY system homologs (Figure 2). Within the phylum Bacillota, two major branchings belonging to the Lactobacillales and the Bacillales orders were included. All Lactobacillales harbored CopY system homologs, one of which harbored both CopY and CsoR homologs (Figure 3). CsoR system homologs were found in all organisms from the Bacillales order, three of which also possessed CopY homologs (Figure 3).

All species from Bacillota harbored the CusRS system. However, Lactobacillales showed fewer hits to the CusRS system (3–8 hits) than Bacillales (7–19 hits). *Enterococcus faecium* EnGen0147 strain A17 Sv1, where the copper-resistant phenotype was first identified (bacterial strain exposed to large amounts of copper), showed a larger CusRS homolog hit count than *E. faecium* strains isolated from the marine sediment (Figure 4). Bacteria from phylum Bacillota showed variability in the hits to the CorE system homologs. *Oceanobacillus* and *Halobacillus* species harbored 9 CorE family hits, while *Metalibacillus* sp. and *Bacillus* sp. harbored up to 16 hits. However, six bacteria from Bacillota showed a single CorE hit. The presence of a few alternate σ factor CorE hits in these organisms might indicate that these organisms do not undergo extreme environmental fluctuation.

Some bacteria belonging to Pseudomonadota, Lentisphaerota, Planctomycetota, and Bacillota phyla also showed homologs of the BlaR family regulator, which controls the abundance of CopY family regulator (PetR) (Figure 2). This system was first characterized in oxygenic photosynthetic Cyanobacteria involved in mediating a switch between iron-containing Cytochrome c6 to copper-containing plastocyanin as an alternative electron carrier [55]. The role of BlaR in these heterotrophic marine species (*Metabacillus* sp., *Oceanobacillus* sp., *Lentisphaera* sp., *Rhodopirullula* sp., *Alteromonas* sp., and *Xanthomonas* sp.) remains to be elucidated.

No hits to the BxmR system belonging to the ArsR/SmtB family were observed, so it was not included in any figures.

Overall, our analyses showed that the distribution of copper-associated signaling system homologs in marine bacteria is mainly explained by phylogenetic classification. However, many copper signaling system hits were widely variable within the same phylum and depict an influence of the environment on the distribution of copper-associated signaling systems.

### 3.2. Potential Role in Host–Pathogen Interaction of the One-Component Signal Transduction System ComR

In our study, phylum Actinomycetota includes marine pathogenic and nonpathogenic strains. Among them, putative ComR homologs were only identified in the pathogenic Actinomycetota (Figure 5). Although ComR was initially studied in *E. coli*, we did not observe a ComR homolog in the *E. coli* strains used in this study [67].

ComR regulates the expression of the outer membrane protein ComC, which is involved in regulating bacterial membrane permeability to copper (Mermod et al., 2012). The identification of ComR homologs on fewer marine species, which were either pathogenic or symbionts, raises the question about their role in this environment. A previous study on this regulator has shown that the ComC-like proteins were not found in any Gram-positive organisms based on NCBI BLAST search analyses [67]. In our analyses of the genomic neighborhood around the ComR homologs, we identified a putative membrane transporter and multidrug efflux transporter in *Mycobacterium marinovum*, a membrane transporter in *Nocardia seriolae*, and a hypothetical protein in *Renibacterium salmoninarum*. This suggests that a putative role of ComR in these strains is to regulate the activity of a membrane protein that might reduce leakage of copper across the membrane or protect cells from copper-associated toxicity by other mechanisms. 

Studies have shown the induction of the Ctr1 copper importer in activated macrophages [94]. Although a precise mechanism for copper-mediated immune response is unknown, the accumulation of copper observed in the phagosome suggests that the innate immune response uses the toxic properties of copper to attack invading pathogens [93]. Copper-mediated defense by the host immune system would exert pressure on pathogenic bacteria, and the membrane-permeabilizing defense activity by the bacteria may be activated under such conditions [95]. Previous studies have also shown virulence-associated copper resistance in human pathogens such as *M. tuberculosis*, where genes encoding copper resistance, specifically P-type ATPase, were upregulated during macrophage infection [96]. These studies suggest that the host’s phagocytic activity uses copper to kill the pathogen-causing infection [97]. In our analyses, the three intracellular marine pathogens of fish that cause a chronic persistent infection were all found to have a ComR homolog not observed in the extracellular pathogens affecting the central nervous system, *Streptococcus iniae* and *Lactococcus griviae* [98] (Figure 4). 

### 3.3. ECF σ Factor CorE in Marine Bacteria and Its Potential Role

Our results show that CorE, an ECF σ factor protein, showed a large variability in the number of putative hits between different organisms. ECF σ factor CorE is the bacterial copper signaling system that helps them adapt to the change in the environmental copper concentration. We observed great variability in the number of CorE homologs found among the marine species (Figure 6). *Bacillus* sp. isolated from sediment and biofilm showed up to 16 CorE hits. In contrast, *Lactobacillus* sp. showed only one hit to CorE (Figure 3). A large homolog hit to the CorE system was observed in organisms found in the biofilm and sediment samples, while pathogenic strains and seawater showed an overall low CorE homolog (Figure 6). The pathogenic strains might only undergo a few environmental fluctuations compared to the marine sediment and biofilm. Based on these observations, the source of isolation plays an essential role in the CorE system distribution. 

Overall, we observed diverse copper-associated signal transduction system homologs belonging to different systems across marine bacteria. All these systems harbored by bacteria found in different marine habitats play an important role in the survival of the bacteria in that environment. It is of essence to characterize the role of different systems under different environmental pressures to explore how these systems can be targeted for novel biotechnological purposes.

## Figures and Tables

**Figure 1 microorganisms-11-01012-f001:**
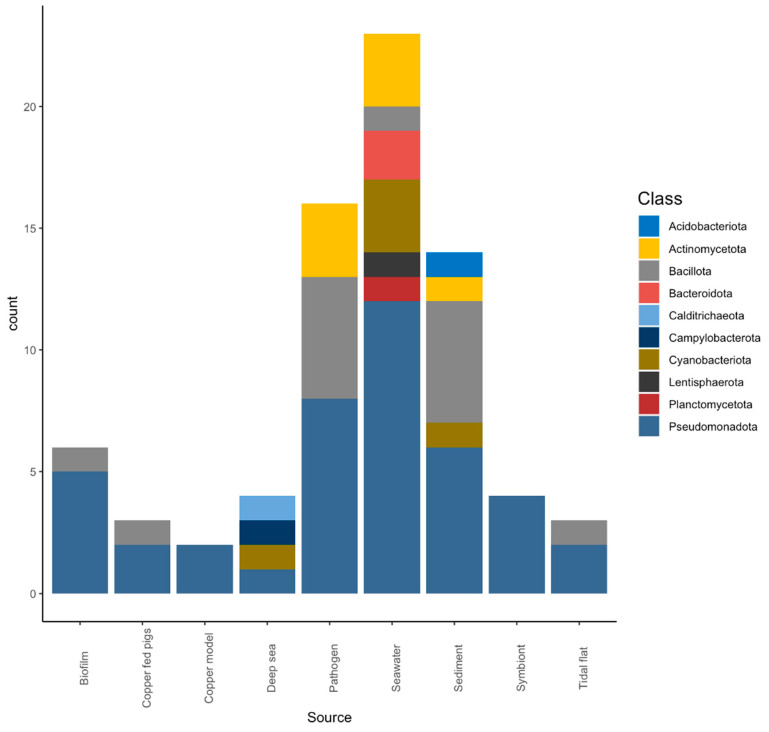
Total number of marine bacteria classified by the phylogenetic system and their source of isolation. X-axis represents isolation source, and Y-axis represents the number of bacterial strains from each source. The colors in the stack plot represent the phylogenetic characterization of the bacteria.

**Figure 2 microorganisms-11-01012-f002:**
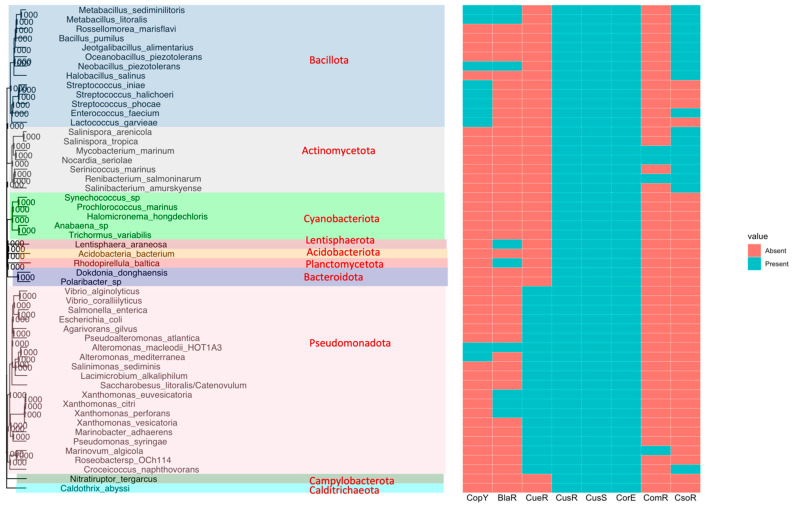
Distribution of copper-associated regulatory systems in marine bacteria from various phylogenetic groups. The nucleotide sequence of the full-length 16S ribosomal RNA sequence of representative marine bacteria was downloaded from the NCBI database. Multiple sequence alignment and phylogenetic analysis (bootstrap of 1000 by neighbor-joining algorithm) were performed in R. The possession of the copper-associated signal transduction system homologs in each organism was mapped onto the phylogenetic tree by using ggplot2.

**Figure 3 microorganisms-11-01012-f003:**
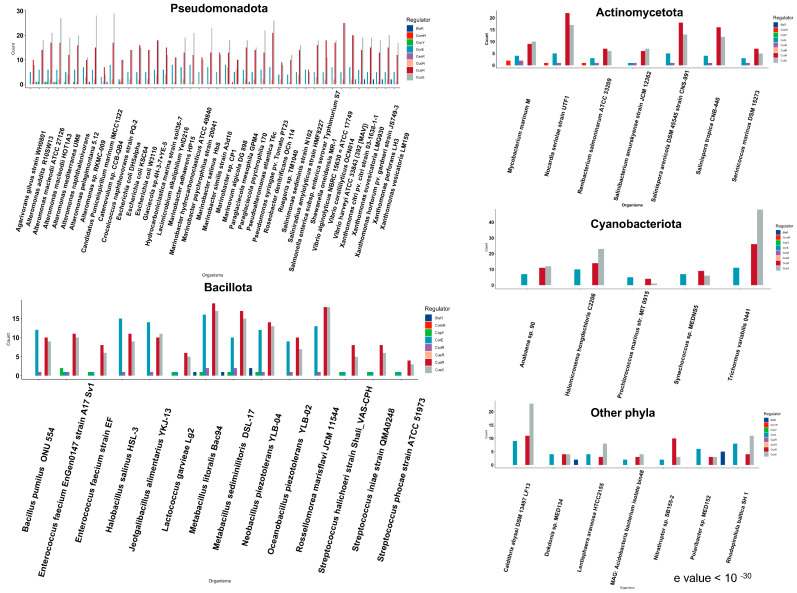
Number of hits to copper-associated signal transduction system in marine bacteria based on phylogeny. The copper-associated system homolog hit count was plotted against each bacterial strain, and each bar color represents a different regulatory system. Panels represent the phyla used. The group “Other classes” included organisms from Acidobacteriota, Bacteroidota, Calditrichaeota, Campylobacterota, Lentisphaerota, and Planctomycetota phyla.

**Figure 4 microorganisms-11-01012-f004:**
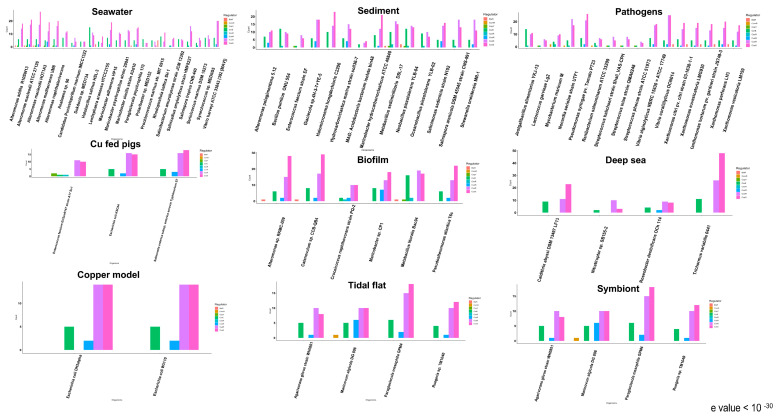
Number of hits to copper-associated signal transduction system in marine bacteria classified according to their isolation source. The copper-associated system homolog hit count was plotted against each bacterial strain, and each bar color represents a different regulatory system. Panels represent bacterial strains classified based on the isolation source.

**Figure 5 microorganisms-11-01012-f005:**
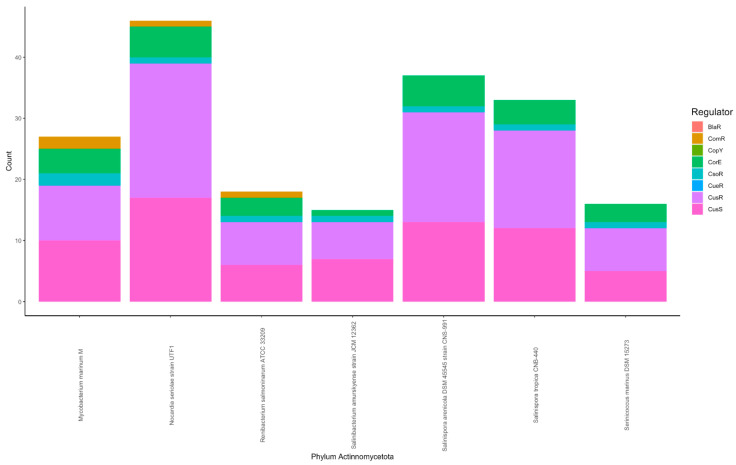
A stacked plot of copper-associated signal transduction system homolog hits in the Actinomycetota. The first three bars in the graph show the pathogenic strains, and the remaining bars show nonpathogenic strains from this group.

**Figure 6 microorganisms-11-01012-f006:**
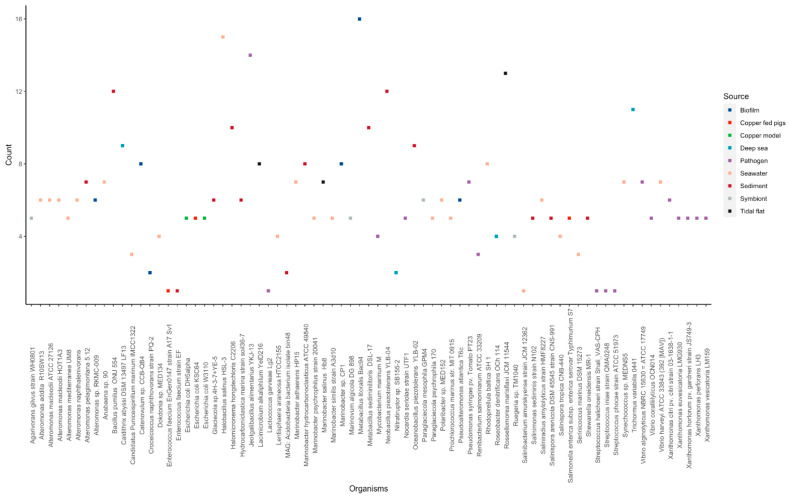
Distribution of the homolog hit count for the CorE system across marine bacteria. Each dot represents the total putative CorE hits for a specific bacterium. The dot colors depict the source of isolation shown in the panel on the left.

**Table 1 microorganisms-11-01012-t001:** Copper-associated protein models used for homology search.

HMM Protein ID	System	Species	NCBI Accession Number
TIGR02044	CueR	*E. coli*	NP_415020.1
TIG02698	CopY	*Enterococcus hirae*	CAA86835.1
PF02583	CsoR	*M. tuberculosis*	NP_215482.1
PF01022	BxmR	*Oscillatoria brevis*	BAD11074.1
PF00440	ComR	*E. coli*	WP_001336528
TIGR02937	CorE	*Myxococcus xanthus*	ABF90558.1
TIGR01386	CusS	*E. coli*	NP_415102.1
	PcoS	*E. coli* Plasmid	CAA58530.1
	CopS	*P. syringae*	QBZ81958.1
TIGR01387	CusR	*E. coli*	NP_415103.1
	PcoR	*E. coli* Plasmid	CAA58529.1
	CopR	*P. syringae*	QBZ81959.1
PTHR34978	BlaR	*Synechocystis* sp.	BAA16703.1

## Data Availability

The data presented in this study are available on request from the corresponding author.

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
