# Peer review of "Linking Copper-Associated Signal Transduction Systems with Their Environment in Marine Bacteria"

_microorganisms, 2023, doi:10.3390/microorganisms11041012_

Round 1

Reviewer 1 Report

Microorganisms in the world has evolved to adapt to their environment, resulting in a unique trait of resistance to harsh environment. Only with a comprehensive understanding of these stress-resistance mechanism of microorganisms, can people develop some novel microorganisms with special functions, and establish corresponding regulation strategies and industrialization technologies. So as to make contributions to the development of human society. In this paper, the existing copper-related signal transduction systems in marine microorganisms are reviewed, and the environmental effects on copper-related signal transduction systems are described in detail. The review can provide the readers meaningful information.

1. The abstract of the paper does not accurately describe the main content of the paper, so it needs to be rewritten.

2. The logic between the paragraphs in the text is not strong; The links between the parts need to be strengthened

 3.The article structure is too wordy, and there is no comprehensive discussion throughout the texts.

Author Response

Reviewer 1:

  1. The abstract of the paper does not accurately describe the main content of the paper, so it needs to be rewritten.

We thank the reviewer for pointing out this concern. We made a significant revision to the abstract to summarize the main contents of the manuscript. The abstract now summarizes the use of copper in marine systems and the presence of different copper-associated regulatory systems across bacteria. We also mention using a comparative genomics study to characterize the presence of different copper signal transduction systems across marine bacteria. We highlight key observations made from the study in the revised abstract.

  1. The logic between the paragraphs in the text is not strong; The links between the parts need to be strengthened.

We agree with the reviewer that the overall organization of the manuscript could be revised to highlight key observations. We made many revisions throughout the paper to clarify this concern. We revised the overall structure in the introduction and the results/discussion section of the manuscript. The introduction of different signal transduction systems now encompasses: 1. Introduction of the system and different genes regulated by that signaling systems, known homologs in other organisms, and signal sensing mechanisms that make each system unique. The results from the comparative analysis are now summarized in sections based on the phylum and we discuss the environmental influence within each phylum. The results for each phylum are now separated as a separate section to highlight key observations made in the analysis. The homology-based nature of the analysis of copper systems across many marine bacteria makes it difficult to summarize every observation in writing for the readers. Still, readers will be able to see those details by referring to Figures 3 and 4.

3.The article structure is too wordy, and there is no comprehensive discussion throughout the texts.

We modified the introduction and results/discussion sections to bring clarity and thoughtfulness, and to cut out some repetitive details from the results/discussion section. This includes a reduction in the introduction on the structural components of the different systems. For the discussion section, we revised the two sections discussing the overall distribution of the signal transduction system and the influence of the environment in a single section. This section focuses on the overall observation of the influence of the environment on the distribution of signal transduction systems across marine bacteria belonging to different groups.

Reviewer 2 Report

This is a thorough survey of the occurrence of copper associated signal transduction systems in marine bacteria. It makes a useful contribution to the field. I have a few comments:

1. The title references "microbes" but "bacteria" would be much more appropriate. I can't see any consideration of eukaryotic microbes or archaea in the paper.

2. The final 2 sentences of the Abstract (lines 22-26) are very cryptic and unhelpful. It would be much better to give a summary of the actual (rather limited) indications that the authors found for association between copper-associated signal transduction and specific lifestyles.

3. I noticed one major omission from the discussion, which is the PetR/PetP (or BlaR/CopY) system recently characterised in a cyanobacterium, in which a copper-responsive membrane protease controls the abundance of a CopY-type response regulator to enable a switch between the expression of plastocyanin and cytochrome c6 (Garcia-Canas et al PNAS 118 (2021). The system seems to be absent from the small number of representative cyanobacteria considered here, but there are homologs in some ecologically-significant marine species, including the nitrogen-fixers Crocosphaera watsonii and Trichodesmium erythraeum. The paper discusses CopY, but it would be interesting to see if there are BlaR homologs in other marine bacteria.

Author Response

Reviewer 2:

  1. The title references "microbes" but "bacteria" would be much more appropriate. I can't see any consideration of eukaryotic microbes or archaea in the paper.

We thank the reviewer for this suggestion. We change the title to refer to bacteria and not microbes.

  1. The final 2 sentences of the Abstract (lines 22-26) are very cryptic and unhelpful. It would be much better to give a summary of the actual (rather limited) indications that the authors found for association between copper-associated signal transduction and specific lifestyles.

We agree with the reviewer about this suggestion. I modified the final 2 sentences of the abstract to highlight observations from the analyses. We, in fact, made an overall correction in the abstract as suggested by both reviewers. As suggested, the final quarter of the abstract now summarizes the use of a comparative genomics study to characterize the presence of different copper signal transduction systems across marine bacteria. It highlights key observations made from the study.

  1. I noticed one major omission from the discussion, which is the PetR/PetP (or BlaR/CopY) system recently characterized in a cyanobacterium, in which a copper-responsive membrane protease controls the abundance of a CopY-type response regulator to enable a switch between the expression of plastocyanin and cytochrome c6 (Garcia-Canas et al PNAS 118(2021). The system seems to be absent from the small number of representative cyanobacteria considered here, but there are homologs in some ecologically-significant marine species, including the nitrogen-fixers Crocosphaera watsonii and Trichodesmium erythraeum. The paper discusses CopY, but it would be interesting to see if there are BlaR homologs in other marine bacteria.

The reviewer makes an excellent suggestion about adding the BlaR system homologs to the analysis. We originally only included the copper-associated signal transduction system that regulates the expression of copper homeostasis systems in our analyses. The PetR/PetP system first identified in Cyanobacteria, regulates the activity of CopY system regulator by copper-responsive BlaR protease, is intriguing. Its presence in many Cyanobacteria, a major component in marine systems, necessitates its inclusion in the analyses. This system is unique, where a membrane protease regulates the cytoplasmic CopY system regulator in response to copper, which is different from CopY’s typical regulation. This would be an excellent addition to the overall analyses of copper-associated systems in marine bacteria and was thus included in the study.

We added two additional marine Cyanobacteriota to the analysis to increase representation from the Cyanobacteriota phylum. However, we did not find the CopY/BlaR homologs in these bacterial species from the Cyanobacteriota phylum. We made some exciting observations of the presence of the BlaR homologs in other heterotrophic marine organisms.

Round 2

Reviewer 1 Report

As it has been revised. This manuscript could be accepted.